# The Cytotoxic Potential of Humanized γδ T Cells Against Human Cancer Cell Lines in *In Vitro*

**DOI:** 10.3390/cells14151197

**Published:** 2025-08-04

**Authors:** Husheem Michael, Abigail T. Lenihan, Mikaela M. Vallas, Gene W. Weng, Jonathan Barber, Wei He, Ellen Chen, Paul Sheiffele, Wei Weng

**Affiliations:** InGenious Targeting Laboratory, 760-81 Coates Avenue, Holbrook, NY 11741, USA; alenihan@genetargeting.com (A.T.L.); mvallas@genetargeting.com (M.M.V.); gwweng26@gmail.com (G.W.W.); jbarber@genetargeting.com (J.B.); whe@genetargeting.com (W.H.); echen@genetargeting.com (E.C.); psheiffele@genetargeting.com (P.S.)

**Keywords:** humanized T cell receptor, γδ T cells, human cancer cells, cytotoxicity, bacterial artificial chromosome genomic DNA platform

## Abstract

Cancer is a major global health issue, with rising incidence rates highlighting the urgent need for more effective treatments. Despite advances in cancer therapy, challenges such as adverse effects and limitations of existing treatments remain. Immunotherapy, which harnesses the body’s immune system to target cancer cells, offers promising solutions. Gamma delta (γδ) T cells are noteworthy due to their potent ability to kill various cancer cells without needing conventional antigen presentation. Recent studies have focused on the role of γδ T cells in α-galactosylceramide (α-GalCer)-mediated immunity, opening new possibilities for cancer immunotherapy. We engineered humanized T cell receptor (HuTCR)-T1 γδ mice by replacing mouse sequences with human counterparts. This study investigates the cytotoxic activity of humanized γδ T cells against several human cancer cell lines (A431, HT-29, K562, and Daudi) *in vitro*, aiming to elucidate mechanisms underlying their anticancer efficacy. Human cancer cells were co-cultured with humanized γδ T cells, with and without α-GalCer, for 24 h. The humanized γδ T cells showed enhanced cytotoxicity across all tested cancer cell lines compared to wild-type γδ T cells. Additionally, γδ T cells from HuTCR-T1 mice exhibited higher levels of anticancer cytokines (IFN-γ, TNF-α, and IL-17) and Granzyme B, indicating their potential as potent mediators of anticancer immune responses. Blocking γδ T cells’ cytotoxicity confirmed their γδ-mediated function. These findings represent a significant step in preclinical development of γδ T cell-based cancer immunotherapies, providing insights into their mechanisms of action, optimization of therapeutic strategies, and identification of predictive biomarkers for clinical application.

## 1. Introduction

Gamma delta (γδ) T cells are a subset of unconventional T lymphocytes that express T cell receptors (TCRs) composed of γ and δ chains. The more common alpha-beta (αβ) T cells, which recognize peptide antigens, are presented by major histocompatibility complex (MHC) molecules. In contrast, γδ T cells can recognize a diverse range of antigens, including non-peptide molecules, and can act independently of the MHC complex. γδ T cells exhibit a distinctive function in immune surveillance and response [1,2,3] and are crucial for both innate and adaptive immunity [4,5,6]. Numerous studies have demonstrated their anticancer effects on renal cell carcinoma, melanoma, colon cancer, breast cancer, and pancreatic ductal adenocarcinoma [7,8,9,10,11,12]. Therefore, γδ T cells represent a promising avenue in cancer immunotherapy.

Alpha-galactosylceramide (α-GalCer) is a synthetic glycolipid and potent immunostimulant. Although α-GalCer is classically recognized by invariant NKT cells through CD1d presentation, emerging studies suggest that γδ T cells may also respond to lipid antigens in certain immunological contexts [13,14]. There is growing interest in exploring whether γδ T cells expressing TCR can directly bind CD1d–lipid complexes or become activated via lipid antigen presentation, including α-GalCer, particularly in the tumor microenvironment (TME) [14,15,16,17].

This binding occurs in a unique orientation over the CD1d A′ pocket, predominantly mediated by the Vδ1 chain, supporting the concept of direct γδ TCR recognition of lipid antigens [14]. Furthermore, in murine models, α-GalCer has been shown to enhance γδ T cell cytotoxicity characterized by increased expression of NKG2D, FasL, CD107a, and Granzyme B. Notably, γδ T cells are essential for achieving full antitumor efficacy in α-GalCer–based immunization models [14].

Notably, α-GalCer demonstrated strong anticancer activity against B16 melanoma-bearing mice [18]. Thus, the application of humanized γδ T cells in α-GalCer- based therapies holds a promise for advancing anticancer treatments.

However, the direct effects of humanized γδ T cells on the activation of cancer cells themselves were not investigated. Understanding these interactions in greater detail could provide new insights into optimizing γδ T cell-based therapies and overcoming resistance mechanisms in cancer treatment.

Through the use of bacterial artificial chromosomes (BACs), our laboratory replaced a sequence of >100 kb to generate humanized γδ mice. This model represents a true gene replacement where the mouse sequence is eliminated and replaced with a human genomic sequence [19].

To investigate the role of humanized γδ T cells in protective immune responses against cancer cells, various human cancer cell lines were co-cultured with humanized γδ T cells and their activity was compared to that of wild-type (WT) mice. Our findings demonstrate that γδ T cells play a critical role in anticancer immunity by stimulating anticancer cytokines that subsequently regulate cytotoxicity activity.

While γδ T cells are known for their cytotoxic potential against tumors, our study is the first to assess this function using γδ T cells derived from humanized γδ TCR (γδ HuTCR-T1) mice and to compare it to that of WT mice γδ T cells, using a novel *in vivo* model engineered to express fully human γδ TCRs. This provides a physiologically relevant system that closely mimics human γδ T cell biology, and this is the first study to explore the role of humanized γδ T cells in human cancer cells’ responses under α-GalCer stimulation.

To contextualize our study, it is important to note that humanized γδ TCR mouse models remain rare and represent a frontier in immunological research. While humanized mice such as those engrafted with human hematopoietic cells (e.g., NSG or TruHuX models) can reconstitute aspects of adaptive immunity, none so far have replaced the mouse γ and δ TCR loci with human equivalents. Our inGenious TruHumanization platform achieves this precisely, enabling the first fully human γδ TCR repertoire in a mouse model, thus providing a unique opportunity to study human-like γδ T cell biology *in vivo*.

## 2. Materials and Methods

### 2.1. Generation of Humanized Gamma Delta (γδ) Humanized T Cell Receptor (HuTCR)-T1 Mice

All animal experiments were conducted in accordance with the Institutional Animal Care and Use Committee (IACUC) protocol (# 2023–1007) approved by the inGenious protocol committee and adhered to the guidelines of the American Veterinary Medical Association and regulations. All the methods are conducted in accordance with ARRIVE guidelines. Animals were housed in groups of three to four per cage under specific pathogen-free conditions with unrestricted access to food and water. HuTCR-T1 γδ mice were generated in our laboratory utilizing the Ingenious TruHumanization platform to incorporate a large bacterial artificial chromosome (BAC) genomic DNA construct (>100 kb) [20]. Humanization was confirmed by genotyping and approximately 6–8-week-old male and female mice were used for these investigations.

### 2.2. Isolation of Mononuclear Cells (MNCs) and Gamma-Delta (γδ) T Cells

At the end of the experimentation, the HuTCR-T1 γδ (C57/129 strain, *n* = 12) and WT (C57 strain, *n* = 12) [20] mice were humanely euthanized using carbon dioxide inhalation. The spleen, lung, lymph nodes, blood, thymus, liver, and ileum were harvested from mice and single cell suspensions were prepared as described previously [19,21,22]. TCR γδ^+^ T cells were isolated from tissues following the manufacturer’s instructions (Miltenyi Biotech, Gaithersburg, MD, USA, cat. # 130-092-125).

### 2.3. Cell Lines and Co-Culture

Mycoplasma free human cancer cell lines HT-29 (colorectal adenocarcinoma, ATCC # HTB-38), A431 (skin epidermoid carcinoma, ATCC # CRL-1555), K562 (chronic myelogenous leukemia, ATCC # CCL-243), and Daudi cells (Burkitt’s lymphoma, ATCC # CCL-213) were cultured in RPMI supplemented with 10% FBS and 1% penicillin and streptomycin (Gibco, Grand Island, NY, USA cat. # 15140-122). We selected A431, HT-29, K562, and Daudi cell lines to represent a diverse range of solid tumors and hematological malignancies. A431 and HT-29 model epithelial cancers, while K562 and Daudi represent myeloid and B-cell lineages. These well-characterized lines differ in MHC expression and stress ligand profiles, making them suitable for assessing the broad cytotoxic capacity of humanized γδ T cells. In addition, these cell lines are known to express CD1d while γδ TCR binds the CD1d-α-GalCer complex [14,17,23,24,25,26,27].

Adherent cells were detached using 0.05% Trypsin-EDTA. The cancer cells (target cells, T) were co-cultured with γδ^+^ T cells (effector cells, E) at the ratio of T/E = 10:1 in 96-well cell culture plates with and without α-GalCer (Cayman Chemical, Ann Arbor, MI, USA, cat. # 24862, 100 ng/mL) for 24 h at 37 °C with 5% CO_2_, and 95% humidity [28]. The ratio 10:1 was chosen based on previously published studies demonstrating optimal cytotoxic responses by γδ T cells when tested against tumor cell lines *in vitro* [29,30,31]. Cell culture media were collected and stored at −20 °C to perform cytotoxicity/ELISA assays while cells were processed for ELISPOT assays.

To block the γδ^+^ T cells’ function, a soluble CD3e blocking peptide (mybiosource.com, San Diego, CA, USA, accessed on 10 October 2023, catalog # MBS8544433) was used at the concentration of 1 μg/mL and added at the beginning of the co-culture.

### 2.4. Lactate Dehydrogenase (LDH) Cytotoxicity and Apoptosis Assay

The lactate dehydrogenase (LDH) cytotoxicity assay is a commonly used method to measure cell cytotoxicity by quantifying the release of LDH, an enzyme presents in the cytoplasm of cells, into the culture medium as a result of cellular membrane rupture or damage. The assay was performed following the manufacturer’s instructions. LDH activity was measured by adding an equal volume of LDH-Glo (Promega, Madison, WI, USA, cat. # J2381) to cell culture media in an opaque plate followed by 30 min incubation at RT. Relative luminescence (RLU) was determined using the Fluoroskan Ascent FL (Thermo Scientific, Bohemia, NY, USA). The cytotoxicity (%) was then calculated as follows:$$\mathrm{Cytotoxicity}\;(\%)\;=\;\frac{\left(\mathrm{Experimental LDH release}-\mathrm{medium background}\right)}{\left(\mathrm{Maximum LDH release in control}-\mathrm{medium background}\right)}$$

### 2.5. Propidium Iodide (PI) Staining for Cell Death

Cell death rate was assessed using the propidium iodide (PI) staining method (Stem Cell, BC, Canada, cat. # 75002 [32]. Co-cultured cells were harvested, washed with the PBS, and resuspended in a solution containing PI (1 µg/mL), and cells were incubated for 15 min at RT protected from light. After the incubation period, stained cells were quantified using a Novocyte^®^ Flow Cytometer (ACEA Biosciences Inc, San Diego, CA, USA) equipped with appropriate fluorescence detectors for PI (488 nm laser for excitation and detecting emission in the FL2/FL3 red channel). A total of 50,000 events were acquired per sample and data were analyzed using Novo Express^®^ software (version 1.6.3).

### 2.6. Enzyme-Linked Immunosorbent Assay (ELISA) and Enzyme-Linked Immunosorbent Spot (ELISPOT) Assay

Murine cytokines (IL-6, IL-17A/F, IFN-γ, TNFα), and Granzyme B were quantified from cell culture supernatants using Invitrogen (Milwaukee, WI, USA) ELISA kits, according to the manufacturer’s instructions. These assays were performed to evaluate cytokines and Granzyme B levels in response to cancer and γδ T cell treatments in various tissues. For cytokine-specific ELISPOT assays, procedures mirrored those of the ELISA assay. Before initiating co-culture, 96-well cell culture plates were coated with the respective cytokine’s capture antibody. After the addition of cancer and γδ T cells, plates were centrifuged for 5 min at 500× RPM and cultured for 24 h in cell culture media. Media were collected and stored at −20 °C for later analysis. Post-culture, plates were washed with PBS, and cytokine-secreting cells (ELISPOTs) were enumerated using an ELISA kit and the tetramethylbenzidine peroxidase substrate system (KPL, cat. # 5120-0053) containing a membrane enhancer (KPL/Sera care, Gaithersburg, MD, USA, cat. # 5420-0026). Enumeration was conducted under a light microscope (Nikon diaphot, CA USA).

### 2.7. Statistical Analysis

Statistical analysis was performed using GraphPad Prism 8.0.2 (GraphPad Software, Inc., San Diego, CA, USA). Data are expressed as the mean ± SEM from four independent experiments having twelve animals per group. Differences between the two groups were analyzed by two-way ANOVA followed by Bonferroni post-test. A *p*-value < 0.05 was considered statistically significant.

## 3. Results

### 3.1. Humanized T Cell Receptor (HuTCR)-T1 Mice Gamma Delta (γδ) T Cells Demonstrated Greater Cytotoxicity Against Human Cancer Cells

Cytotoxicity was measured by LDH levels in the cell culture media, a reliable indicator of cell death associated with cell damage and rupture. Co-culturing various human cancer cell lines with γδ T cells for 24 h in the presence of α-GalCer consistently demonstrated that HuTCR-T1 γδ T cells exerted a significantly greater cytotoxic effect across most tested cancer cell lines and tissues compared to WT γδ T cells, with the most pronounced effect observed in Daudi cells (Figure 1).

PI staining was used to perform a cell death assay on co-cultured cells. Nonviable cells exhibited increased PI fluorescence in comparison to viable cells. The data in Figure 2 indicate that in most cases HuTCR-T1 γδ T cells have significantly increased cell death compared to WT γδ T cells across most tissues and cancer cell lines tested. The observed increase in cell death rate suggests that HuTCR-T1 γδ T cells have enhanced cytotoxic activity compared to WT γδ T cells, further emphasizing their potential as effective mediators of anticancer immune responses.

Co-culture experiments were also performed in the absence of α-GalCer supplementation, demonstrating similar trends with a lesser degree of cytotoxic activity (Appendix A). Taken together, HuTCR-T1 γδ T blood cells exhibited significant cytotoxicity in comparison to WT γδ T blood cells in all cancer cell lines and supported the enhanced cytotoxic potential of HuTCR-T1 γδ T cells against human cancer cells. The data strongly suggest that HuTCR-T1 γδ T cells’ superior cytotoxic effects may be due to γδ TCR receptors with a heightened affinity for human cancer cells.

### 3.2. HuTCR-T1 Mice γδ T Cells Enhanced the Production of Anticancer Cytokine-Specific ELISPOTs in Response to Stimulation

The ELISPOT assay is a highly sensitive immunoassay, allowing for the measurement of cytokine-secreting cells at the single-cell level. Here, the ELISPOT assay was used to detect cytokine-specific producing cells. Cell culture plates were coated with respective capture antibodies before initiation of co-culture.

Data revealed that the mean numbers of IFN-γ/TNFα-specific ELISPOTs were significantly greater (in most cases) in HuTCR-T1 γδ T cells compared to WT γδ T cells across all tested cancer cell lines and most tissues (Table 1). However, in liver samples from the A431 and HT-29 cell lines, the mean numbers of TNFα-specific ELISPOTs were slightly higher in WT γδ T cells compared to HuTCR-T1 γδ T cells. The mean number of IL-17-specific ELISPOTs was also increased in most cancer cell lines and tissues when comparing HuTCR-T1 γδ T cells to WT γδ T cells. Daudi and HT-29 cells had significantly higher IL-17-specific ELISPOTS in liver cells for HuTCR-T1 cells compared to WT. The significant increase was also seen in splenic and ileal cells for A431 cells (Table 1).

Conversely, IL-17-specific ELISPOTs were slightly elevated in WT mice compared to HuTCR-T1 γδ T cells in spleen, blood, liver, and ileum when co-cultured with K562 cells; in lungs when co-cultured with A431 cells; and in blood when co-cultured with HT-29 cells (Table 1). Of key importance, IL-6-specific ELISPOTs were undetectable in our co-culture experiments across all tested cancer cell lines and tissues, suggesting that IL-6 production may not play a significant role in the interaction between γδ T cells and cancer cells under the conditions tested.

Overall, our findings demonstrate that HuTCR-T1 γδ T cells exhibit enhanced anticancer activity compared to WT γδ T cells when exposed to a stimulant.

### 3.3. HuTCR-T1 Mice γδ T Cells Increased the Levels of Anticancer Cytokines and Granzyme B in Response to Stimulation

The quantitative ELISA assays employed in our study allowed for the assessment of various anticancer cytokines produced in the cell culture media. Our findings indicated that HuTCR-T1 γδ T cells produced higher levels of IFN-γ and TNF-α compared to WT γδ T cells across most tested cancer cell lines and tissues (Appendix A). However, in some cases, WT γδ T cells showed slightly higher levels of certain cytokines than HuTCR-T1 γδ T cells, particularly IFN-γ in the ileal cells co-cultured with A431 cells, IFN-γ in the blood and liver cells co-cultured with HT-29 cells, and IFN-γ in the splenic cells co-cultured with K562 cells (Appendix A).

Although statistically insignificant, IL-6 levels were numerically higher in HuTCR-T1 γδ T cells compared to WT γδ T cells across most cell lines and tissues. However, exceptions were observed where WT γδ T cells showed higher IL-6 levels, specifically in the lung and liver cells when co-cultured with Daudi cells; splenic and lung cells when co-cultured with A431 cells; splenic and lung cells when co-cultured with HT-29 cells; and thymus, blood, and lung cells when co-cultured with K562 cells (Appendix A).

Furthermore, IL-17 production was marginally higher in WT γδ T cells compared to HuTCR-T1 γδ T cells, although the differences were statistically insignificant. Notably, there were instances where HuTCR-T1 γδ T cells demonstrated numerically higher IL-17 production compared to WT γδ T cells, particularly in thymus and liver cells when co-cultured with Daudi cells and splenic and liver cells when co-cultured with K562 cells (Appendix A).

Furthermore, HuTCR-T1 γδ T cells significantly increased production of Granzyme B compared to WT γδ T cells across all tested cell lines and tissues, highlighting their ability to exert potent anticancer effects through mechanisms such as Granzyme B-mediated cytotoxicity (Appendix A).

Overall, our results demonstrate that HuTCR-T1 γδ T cells enhanced the production of anticancer cytokines and Granzyme B compared to WT γδ T cells, emphasizing their potential as effective mediators of anticancer immune responses.

### 3.4. αCD3e Effectively Abrogated the Functional Effect of γδ T Cells

To confirm the mechanism of inhibition, we employed αCD3e, providing valuable insights into the role of γδ T cells in mediating the observed anticancer effects. When γδ T cells were co-cultured with cancer cells in the presence of α-GalCer and αCD3e blocker for 24 h, the CD3e blocker completely abrogated the effect of the γδ T cells (Figure 3). These results strongly suggest that the observed anticancer effects were mediated by γδ T cells through a directly targeted interaction with cancer cells, which were effectively blocked.

## 4. Discussion

Previous studies have highlighted the critical role of γδ T cells in mediating anticancer effects and regulating metastasis in both murine and human models [7,8,9,10,33]. To date, no animal model has yet been developed that can replicate the diverse array of human γδ TCRs generated through VDJ recombination. To address this gap, we engineered a novel γδ TCR mouse model, termed HuTCR-T1 γδ mice that express human Vγ9-11 and δ1-δ8 TCRs [19,20]. In these mice, highly diverse repertoires of human γδ TCRs were identified. Cytotoxic activities of HuTCR-T1 γδ T cells against various cancer cell lines were assessed to demonstrate the potential for clinical application.

Our data compliment previous findings to confirm that humanized γδ T cells expressing γδ TCR receptors may have a high affinity for human cancer cells and the CD1d-αGalCer complex, generating a robust cytotoxic effect [7,8,9,10,11,12,14,17,20,33]. In addition, we demonstrated that blockage by αCD3e may have disrupted the interaction between γδ T cells and their target cancer cells. This disruption could lead to the downstream inhibition of signaling and effector functions, which are essential for the anticancer activity of the γδ T cells. Taken together, these data indicate that the anticancer effect of our humanized γδ T cells may be specifically mediated via the γδ T cells.

γδ T cells are recognized for their strong cytotoxic activity mediated via the Granzyme–perforin axis [34]. Granzyme B enters target cells through pores created by perforin, leading to apoptosis in the target cells such as cancerous or infected cells [34]. Our observation of increased Granzyme B production suggests that HuTCR-T1 γδ T cells exhibit enhanced cytotoxicity against human cancer cells. This may be further supported by their ability to induce cell death, indicating a potentially greater efficacy in targeting and eliminating cancer cells compared to WT mouse γδ T cells. Additionally (although not investigated), γδ T cells are known to express death ligands such as FasL and TRAIL, which may bind to their respective receptors on cancer cells, thereby triggering apoptosis through an extrinsic pathway [35,36,37,38].

Our findings align with the previous studies demonstrating that γδ T cells produce cytokines, including cytokine-specific IFN-γ, TNF-α, and IL-17 ELISPOTs [7,39,40,41,42,43,44,45,46]. These cytokines collectively contribute to the anticancer immune responses and induce direct cytotoxic effects on cancer cells. Additionally, γδ T cells may also enhance or modulate the TME, which facilitates cancer cells’ recognition [47]. Understanding the roles of these cytokines in cancer immunity could provide valuable insights for researchers developing γδ T cell-based immunotherapies and/or optimizing their use in clinical settings [48].

The dual roles of TNF-α and IL-17 in cancer emphasize the need for a nuanced approach to cancer immunotherapy. Combining immune checkpoint inhibition [49] with strategies targeting TNF-α, IL-17, and γδ T cells presents a promising avenue to enhance antitumor immunity while mitigating pro-tumorigenic effects.

γδ T cells can also produce IL-6, which may significantly influence the TME and impact the immune response to cancer [33,50,51,52,53,54,55]. In our studies, IL-6 levels were marginally elevated in the culture media; however, the absence of detectable IL-6-specific ELISPOTs suggests that IL-6 may not play a central role in γδ T cell-mediated anticancer activity under these experimental conditions. Several factors could explain why IL-6 did not appear critical in our findings: (1) γδ T cells may exert their anticancer effects through alternative cytokines, such as IFN-γ, TNF-α, or IL-17, or via mechanisms independent of IL-6, including direct cytotoxicity; (2) the observed IL-6 in the media might be produced by cancer cells rather than γδ T cells; (3) γδ T cells may transiently produce IL-6, but not at levels sustained long enough for detection by the ELISPOT assay, which captures cytokine production at a specific time point; and (4) IL-6 could be acting in a paracrine manner, independent of direct γδ T cell influence. Further investigation is needed to clarify the role of IL-6 and to elucidate the mechanisms underlying γδ T cell function.

While co-culturing humanized γδ T cells with human cancer cell lines *in vitro* provides valuable insights into the potential therapeutic strategies and the cytotoxic potential of γδ T cells, there are several limitations to consider: (1) non-natural environment: *in vitro* studies do not fully replicate the complex cancer microenvironment found *in vivo*; (2) modified/simplified models: cancer cell lines may often lack the heterogeneity and complexity seen in primary cancers; (3) immune escape mechanisms: cancer cells have evolved various mechanisms to evade immune detection and elimination; and (4) although these mice express human Vγ9Vδ2 TCRs, other components of their immune system, such as cytokine signaling networks, antigen-presenting cells, and stromal architecture, remain murine. These disparities may influence the functional behavior and activation thresholds of humanized γδ T cells, potentially impacting the interpretation and generalizability of the observed immune responses. Therefore, caution should be exercised when extrapolating these findings to human clinical contexts. However, despite these limitations, *in vitro* co-culture systems remain an important tool for investigating the interactions between γδ T cells and cancer cells, highlighting the underlying mechanisms, and screening potential therapeutic interventions.

Studying the cytotoxic potential of humanized γδ T cells against human cancer cell lines *in vitro* offers several advantages. *In vitro* assays provide (1) controlled and reproducible experimental conditions, (2) high-throughput screening, (3) mechanistic insights, (4) comparative analyses, (5) modeling of immune responses, (6) target validation, and (7) ethical considerations. Collectively, these strengths make *in vitro* experiments a valuable tool for bridging the gap between laboratory research and clinical application in cancer immunotherapy.

In future studies, we plan to investigate (1) the effects of α-GalCer on cancer cells in HuTCR-T1 γδ mice *in vivo* to better elucidate therapeutic potential and underlying mechanisms including tumor clearance, survival outcomes, and cytokine responses; (2) the combinatorial effects of γδ T cells with other immunotherapeutic agents; (3) biomarkers predictive of response to γδ T cell-based therapies; and (4) human γδ TCRs that directly interact with human cancer cells. These efforts aim to strengthen our understanding of humanized γδ T cells in cancer therapy and contribute to the development of novel, effective treatments for cancer patients.

In conclusion, our study underscores the enhanced anticancer efficacy of HuTCR-T1 γδ T cells over WT mouse γδ T cells, demonstrating their potential as powerful effectors in cancer immunotherapy. The combination of increased anticancer cytokine production, elevated Granzyme B levels, and effective induction of cytotoxicity/cell death highlights the therapeutic potential of these genetically modified γδ T cells. Future research priority will focus on further elucidating the mechanisms underlying their superior cytotoxicity and exploring *in vivo* and/or clinical applications in various cancer types.

## Figures and Tables

**Figure 1 cells-14-01197-f001:**
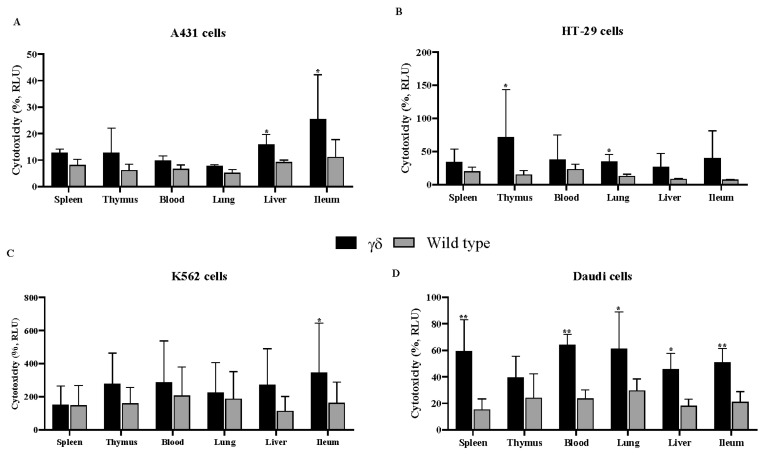
Humanized T cell receptor (HuTCR)-T1 gamma delta (γδ) mice exhibited greater cytotoxic effect. γδ T cells were co-cultured with human cancer cells [A431 (**A**), HT-29 (**B**), K562 (**C**), and Daudi (**D**)] at the ratio of 1:10 in the presence of α-galactosylceramide (α-GalCer) for 24 h. Cytotoxicity assay was performed using lactate dehydrogenase assay. Differences between the two groups were analyzed by two-way ANOVA followed by Bonferroni post-test. Mean numbers from four independent experiments are reported as ± SEM (*n* = 12). Humanized mice γδ T cells were compared with wild-type γδ T cells; * *p* < 0.05, and ** *p* < 0.01 were considered statistically significant.

**Figure 2 cells-14-01197-f002:**
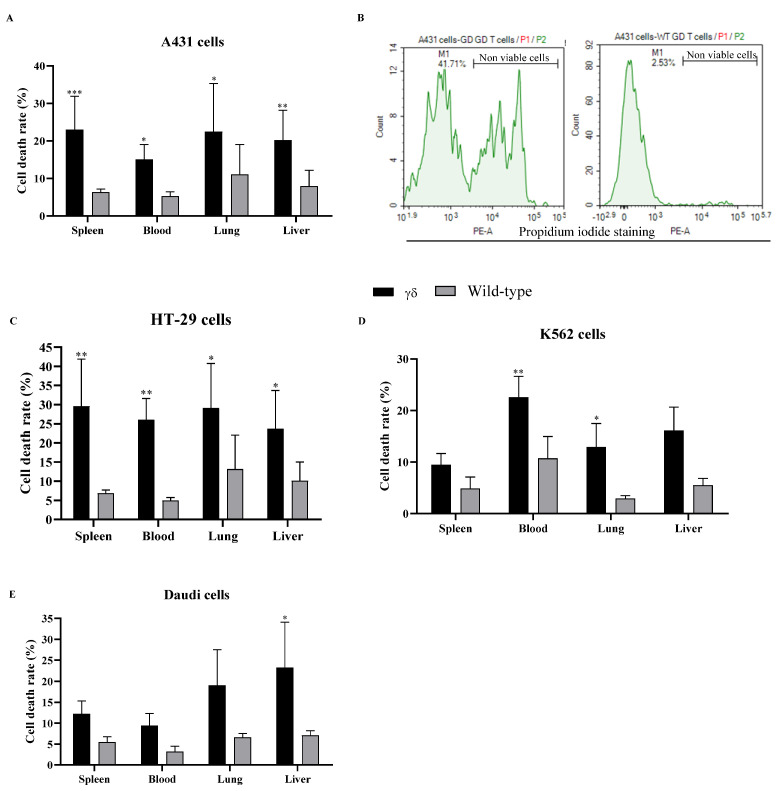
Humanized T cell receptor (HuTCR)-T1 gamma delta (γδ) mice exhibited greater apoptotic effect. γδ T cells were co-cultured with human cancer cells [A431 (**A**,**B**), HT-29 (**C**), K562 (**D**), and Daudi (**E**)] at the ratio of 1:10 in the presence of α-galactosylceramide (α-GalCer) for 24 h. Cell death assay was performed using propidium iodide (PI) staining. Differences between the two groups were analyzed by two-way ANOVA followed by Bonferroni post-test. Mean numbers from four independent experiments are reported as ± SEM (*n* = 12). Humanized mice γδ T cells were compared with wild-type γδ T cells; * *p* < 0.05, ** *p* < 0.01, and *** *p* < 0.001 were considered statistically significant.

**Figure 3 cells-14-01197-f003:**
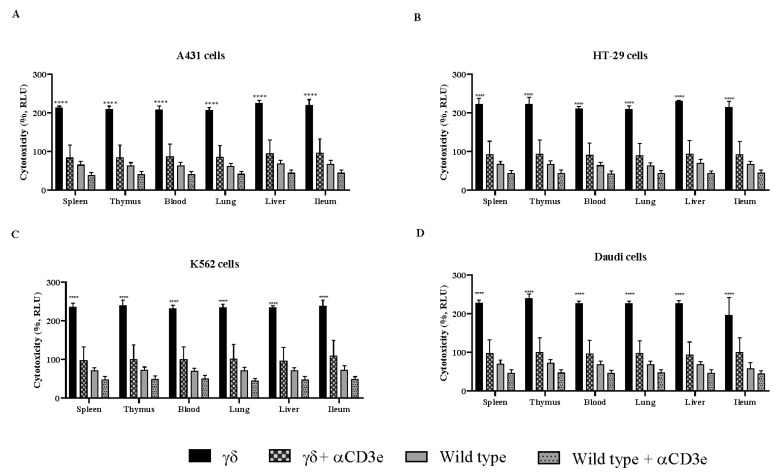
αCD3e abrogated the effect of gamma delta (γδ) T cells. γδ T cells were co-cultured with human cancer cells [A431 (**A**), HT-29 (**B**), K562 (**C**), and Daudi (**D**)] at the ratio of 1:10 in the presence of α-galactosylceramide (α-GalCer) and αCD3e for 24 h. Cytotoxicity assay was performed using the lactate dehydrogenase in the cell culture media. Differences between the two groups were analyzed by two-way ANOVA followed by Bonferroni post-test. Mean numbers from four independent experiments are reported as ± SEM (*n* = 12). Humanized mice γδ T cells were compared with wild-type γδ T cells; **** *p* < 0.0001.

**Table 1 cells-14-01197-t001:** Mean numbers of cytokines specific ELISPOTs.

	Spleen						Thymus						Blood						Lung						Liver						Ileum					
	IFN-γ		TNF-α		IL-17		IFN-γ		TNF-α		IL-17		IFN-γ		TNF-α		IL-17		IFN-γ		TNF-α		IL-17		IFN-γ		TNF-α		IL-17		IFN-γ		TNF-α		IL-17	
Cell line	γδ	WT	γδ	WT	γδ	WT	γδ	WT	γδ	WT	γδ	WT	γδ	WT	γδ	WT	γδ	WT	γδ	WT	γδ	WT	γδ	WT	γδ	WT	γδ	WT	γδ	WT	γδ	WT	γδ	WT	γδ	WT
A431	*** 20	0	** 17	2.7	* 4	0	**** 24	1.6	** 14	2.2	3	0	* 12	1.6	* 12	0.5	2	0	* 15	4.1	** 18	3.8	2	2.6	5	4.4	2	2.6	1	0.2	** 15	1.7	2	0	** 4	0
HT-29	*** 35	2	** 22	2	3	2	* 21	6	* 17	2	4	2	* 20	2	* 13	2	2	3	* 24	4	9	5	4	1	7	1	9	10	* 7	3	8	4	6	4	3	1
K562	**** 21	5	*** 27	10	3	5	* 8	1	4	1	4	1	* 8	1	3	1	1	2	* 10	1	* 9	1	4	2	**** 25	5	** 22	7	2	* 8	**** 20	4	*** 21	5	2	5
Daudi	** 8	0	* 8	0	3	0	4	0	1	0.6	1	0.6	* 5	0.8	* 4	0	3	0.4	* 9	0	** 12	1.3	4	0.7	* 16	6.8	* 13	4.8	* 7	2.1	** 13	2.5	* 11	2.1	5	3.5

Humanized T cell receptor (HuTCR)-Tl gammadelta (γδ) and wild-type (WT) mice, γδ T cells were isolated from various tissues and co-cultured in the presence of α-galactosylceramide for 24 h with A431, HT-29, K562, and Daudi cells at the ratio of 1:10. ELISPOTs were developed with a TMB substrate method containing membrane developer and cytokine specific ELISPOTs were counted using a light microscope (40×). Mean numbers from four independent experiments were reported as ± SEM (n = 12). Humanized mice γδ T cells were compared with WT γδ T cells, * *p* < 0.05, ** *p* < 0.01, *** *p* < 0.001, and **** *p* < 0.0001 were considered as statistically significant. IFN-γ, interferon gamma; TNF-α, tumor necrosis factor alpha; IL-17, interleukin-17.

## Data Availability

The data sets generated during and/or analyzed during the current study are available from the corresponding author upon a reasonable request.

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
