# Peer review of "The Cytotoxic Potential of Humanized γδ T Cells Against Human Cancer Cell Lines in In Vitro"

_cells, 2025, doi:10.3390/cells14151197_

Round 1
Reviewer 1 Report
Comments and Suggestions for Authors
The paper focuses on the potential of humanized γδ T cells in cancer immunotherapy. By constructing the HuTCR-T1 humanized T cell receptor mouse model, the study investigates the cytotoxicity of γδ T cells against human cancer cell lines such as A431, HT-29, K562, and Daudi in vitro. The findings reveal that compared to wild-type human γδ T cells, the humanized γδ T cells exhibit stronger cytotoxicity under both α-GalCer - stimulated and unstimulated conditions. They induce more cancer cell apoptosis and secrete higher levels of anticancer cytokines (IFN-γ, TNF-α, IL-17) and granzyme B. Their function can be blocked by αCD3e, confirming the γδ T cell - mediated effects. Overall, the study provides experimental evidence for the preclinical development of γδ T cell - based cancer immunotherapy. However, there are several issues that need to be addressed before publication.
- The introduction lacks a comprehensive review of the existing research progress on humanized γδ T cell receptor models. It does not clearly highlight the differences and innovations of this study compared to existing models. It is recommended to supplement relevant background information to emphasize the research value. Additionally, the description of the mechanism of action of α-GalCer in γδ T cells is not in-depth enough. It is suggested to add more literature support on how α-GalCer exerts anticancer effects through γδ T cells.
- In the materials and methods section, the generation of HuTCR-T1 mice only mentions the use of BAC technology without detailing the specific length of the humanized sequence, the mouse gene region that was replaced, and the key parameters for BAC construction. This affects the reproducibility of the methodology.
- The cell line cultivation section does not specify whether antibiotics were added to the culture medium, the results of mycoplasma testing for the cell lines, or the authentication status of the cell lines (e.g., STR profiling). These details need to be added to ensure the reliability of the cell lines. The study selected A431, HT-29, K562, and Daudi cell lines but did not fully explain the rationale for choosing these cell lines. It is recommended to supplement information on the background of these cell lines in terms of CD1d expression or sensitivity to γδ T cells.
- The basis for the effector-to-target cell ratio (10:1) in the co - culture experiments is not clear. It is not specified whether a ratio optimization experiment was conducted. It is suggested to add an analysis of the rationality of the ratio selection. To better present the experimental results, it would be helpful to provide images of T cells and tumor cells co - incubated.
- The data presentation in Figures 1 and 2 is relatively clear, but the layout of the tables (e.g., Table 2) is disorganized, making some data difficult to read. It is recommended to redesign the tables to ensure that the data are clear and readable.
- When detecting apoptosis using PI staining, PI alone cannot specifically distinguish between apoptotic and necrotic cells. It is suggested to add methods such as Annexin V staining to improve the specificity of apoptosis detection.
- In the discussion, it is proposed that the enhanced cytotoxicity of HuTCR-T1 γδ T cells is related to the affinity of the γδ TCR, but there is a lack of direct experimental evidence (e.g., affinity measurement data). It is suggested to supplement or adjust the statement to avoid overly conclusive remarks.
- The limitations analysis does not mention the impact of the differences between the in vivo immune microenvironment of HuTCR-T1 mice and that of humans on the clinical translation of the research results. This part of the discussion needs to be added to complete the assessment of research limitations.
- Some sentences are wordy or grammatically unclear (e.g., “Take together” should be “Taken together”). It is recommended to polish the language of the entire text to ensure concise and accurate expression.
- The reference section has inconsistent citation formats (e.g., the notation of PubMed PMID). It is suggested to unify the reference format and ensure that all citations are directly related to the content of the text.

Author Response
Dear Reviewer # 1,
Thank you for the opportunity to respond to the comments on our manuscript. We appreciate the reviewers’ thorough evaluation and constructive feedback, which have helped us improve the quality of our work. We have addressed this point by clarifying details, a method, and revising sections. The relevant changes have been made in the manuscript. We believe these revisions have substantially strengthened the manuscript and hope it now meets the journal’s standards for publication. Thank you again for your time and consideration.
We have carefully addressed each point raised and summarize our responses below.
REVIEWER # 1
- The introduction lacks a comprehensive review of the existing research progress on humanized γδ T cell receptor models. It does not clearly highlight the differences and innovations of this study compared to existing models. It is recommended to supplement relevant background information to emphasize the research value. Additionally, the description of the mechanism of action of α-GalCer in γδ T cells is not in-depth enough. It is suggested to add more literature support on how α-GalCer exerts anticancer effects through γδ T cells.
RE: We sincerely appreciate the reviewer’s valuable feedback regarding the introduction. In response, we have revised the Introduction section to incorporate a more comprehensive overview of existing research on humanized γδ TCR.
‘To contextualize our study, it’s important to note that humanized γδ TCR mouse models remain rare and represent a frontier in immunological research. While humanized mice such as those engrafted with human hematopoietic cells (e.g., NSG or TruHuX models) can reconstitute aspects of adaptive immunity, none so far have replaced the mouse γ and δ TCR loci with human equivalents. Our inGenious TruHumanization platform achieves this precisely, enabling the first fully human γδ TCR repertoire in a mouse model, thus providing a unique opportunity to study ‘human-like γδ T cell biology in vivo’. (Lines 113-119).
‘This binding occurs in a distinctive orientation over the CD1d A′ pocket, mediated primarily by the Vδ1 chain, thereby supporting the notion of direct γδ TCR engagement with lipid antigens. Moreover, in murine systems, α-GalCer can augment γδ T cell cytotoxicity marked by upregulation of NKG2D, FasL, CD107a, and granzyme B and γδ T cells are required for full antitumor efficacy in α-GalCer mediated immunization models’. (Lines 86-91).
- In the materials and methods section, the generation of HuTCR-T1 mice only mentions the use of BAC technology without detailing the specific length of the humanized sequence, the mouse gene region that was replaced, and the key parameters for BAC construction. This affects the reproducibility of the methodology.
RE: The primary focus of this paper is to demonstrate the primary analysis of cytotoxicity of humanized γδ T cells in vitro. We have taken steps to protect this innovative development by submitting a patent application for the HuTCR-T1 mouse model. The patent application details the comprehensive process involved in the creation, validation, and characterization of these mice. The patent application: Weng W. Genetically modified non-human having humanized gamma and delta TCR variable genes. Patent application publication. 2023:US20240114883A1. We have attached a PowerPoint presentation (please see file gamma delta PPT in non published material section) that includes the human sequences used to construct the BAC (> 100Kb) vectors. The composition and verification of gene expression, as well as the confirmation of humanized γδ TCR via flow cytometry can be found on slides 18-23 and slide 24, respectively.
- The cell line cultivation section does not specify whether antibiotics were added to the culture medium, the results of mycoplasma testing for the cell lines, or the authentication status of the cell lines (e.g., STR profiling). These details need to be added to ensure the reliability of the cell lines. The study selected A431, HT-29, K562, and Daudi cell lines but did not fully explain the rationale for choosing these cell lines. It is recommended to supplement information on the background of these cell lines in terms of CD1d expression or sensitivity to γδ T cells.
RE: The authors apologize for the omission regarding the use of antibiotics in the cell culture. 1% penicillin-streptomycin (Gibco 15140-122) was added to all cell culture media. (Line 144)
All cell lines were purchased and certified mycoplasma-free through ATCC. (Line 141).
We selected the A431, HT-29, K562, and Daudi cell lines to represent a diverse spectrum of human solid tumors and hematological malignancies. This panel allows us to assess the broad cytotoxic potential of humanized γδ T cells against both epithelial-derived and lymphoid-derived cancer cells. A431 and HT-29 represent epithelial solid tumors of skin and gut origin, while K562 and Daudi are established models for myeloid and B-cell lineage leukemias/lymphomas, respectively. These cell lines are well-characterized, widely used in immune-oncology studies, and differ in MHC expression, susceptibility to stress ligands, and responsiveness to innate cytotoxicity, making them suitable for evaluating γδ T cell-mediated immune responses. (Lines…145-149).
Moreover, the background of these cell lines in terms of CD1d expression or sensitivity to γδ T cells has been addressed by several references (Line 150).
- The basis for the effector-to-target cell ratio (10:1) in the co-culture experiments is not clear. It is not specified whether a ratio optimization experiment was conducted. It is suggested to add an analysis of the rationality of the ratio selection. To better present the experimental results, it would be helpful to provide images of T cells and tumor cells co-incubated.
RE: We thank the reviewer for highlighting the need to clarify the basis for selecting the 10:1 effector-to-target (E:T) ratio in our co-culture assays. This ratio was chosen based on previously published studies demonstrating optimal cytotoxic responses by γδ T cells when tested against tumor cell lines in vitro and relevant references have been added to the manuscript (Lines…156, ref #s 29, 30, 31).
Regarding the suggestion to include co-culture images, we acknowledge the value of such data for visual confirmation. However, imaging was not performed in this experimental setup, as our assays focused on quantitative cytotoxicity endpoints (e.g., LDH release, ELISPOT). We will consider including live-cell imaging or immunofluorescence in future studies to visualize effector-target interactions more directly.
- The data presentation in Figures 1 and 2 is relatively clear, but the layout of the tables (e.g., Table 2) is disorganized, making some data difficult to read. It is recommended to redesign the tables to ensure that the data are clear and readable.
RE: We thank the reviewer for the comment regarding Table 2. Table 2 has now been reformatted into an Excel file and is included as Supplementary Table 1. Lines…275.
- When detecting apoptosis using PI staining, PI alone cannot specifically distinguish between apoptotic and necrotic cells. It is suggested to add methods such as Annexin V staining to improve the specificity of apoptosis detection.
RE: We thank the reviewer for this insightful suggestion. We acknowledge that PI staining alone cannot reliably distinguish apoptotic from necrotic cells due to its nonspecific DNA intercalation. While our initial approach focused on PI-based detection for general cell death, the term ‘apoptosis’ has been replaced with ‘cell death rate’. At present, we are in the process of expanding and characterizing the HuTCR-T1 γδ mouse colonies. Due to these logistical and technical constraints, we were unable to include new data in the current study. (Lines…227, Figure 2).
- In the discussion, it is proposed that the enhanced cytotoxicity of HuTCR-T1 γδ T cells is related to the affinity of the γδ TCR, but there is a lack of direct experimental evidence (e.g., affinity measurement data). It is suggested to supplement or adjust the statement to avoid overly conclusive remarks.
RE: We thank the reviewer for this important comment. We agree that our current study does not provide direct experimental evidence for affinity measurements to conclusively link enhanced cytotoxicity of HuTCR-T1 γδ T cells to increased γδ TCR affinity. In response, we have revised the discussion by adding ‘may’ to clarify that this proposed mechanism is speculative and based on the observed functional differences between HuTCR-T1 and wild-type γδ T cells. Lines …241, 319,
- The limitations analysis does not mention the impact of the differences between the in vivo immune microenvironment of HuTCR-T1 mice and that of humans on the clinical translation of the research results. This part of the discussion needs to be added to complete the assessment of research limitations.
RE: We appreciate the reviewer’s insightful comment. We agree that differences between the in vivo immune microenvironment of HuTCR-T1 mice and that of humans may influence the clinical translatability of our findings. To address this, we have revised the Discussion section to include this important limitation.
‘Although these mice express human Vγ9Vδ2 TCRs, other components of their immune system, such as cytokine signaling networks, antigen-presenting cells, and stromal architecture, remain murine. These disparities may influence the functional behavior and activation thresholds of humanized γδ T cells, potentially impacting the interpretation and generalizability of the observed immune responses. Therefore, caution should be exercised when extrapolating these findings to human clinical contexts. (Lines 364-369).
- Some sentences are wordy or grammatically unclear (e.g., “Take together” should be “Taken together”). It is recommended to polish the language of the entire text to ensure concise and accurate expression.
RE: We appreciate the reviewer’s helpful comment regarding the clarity and conciseness of the manuscript. We have thoroughly revised the text to improve grammar, eliminate wordiness, and ensure accurate expression throughout the manuscript. (Lines 238, 323).
- The reference section has inconsistent citation formats (e.g., the notation of PubMed PMID). It is suggested to unify the reference format and ensure that all citations are directly related to the content of the text.
RE: We thank the reviewer for pointing out the inconsistencies in the reference formatting. We have thoroughly revised the reference section to ensure a uniform citation style throughout the manuscript.

Reviewer 2 Report
Comments and Suggestions for Authors
The report by Michel and colleagues presents the results from various human cancer cell lines co-cultured with and without α-GalCer, and humanized Humanized T cell receptor (HuTCR)-T1 mice gamma delta (γδ) T cells to explore whether γδ T expressing TCR can recognize CD1d–lipid complexes or become activated via lipid antigen presentation. They used standardized methodologies to verify the γδ T cytotoxic effects and possible biochemical mediators. Figures 1 and 2 show that HuTCR-T1 gamma delta (γδ) from various tissues from the mice exhibited greater cytotoxic effect compared to wild γδ T cells. Table 1 and 2 show that HuTCR-T1 mice γδ T cells after activation can express high the levels of various anticancer cytokines (IFN, TNF, IL-6, IL-17) and Granzyme B. Overall the results show that HuTCR-T1 γδ T mice model could be a good system for exploring the molecular mechanisms and possible clinical benefits of the adoptive transplantation of γδ T cells in patients. These aspects and other important considerations were described in the Discussion section of the report. One of the limitations of this study is that in vivo experiments were not performed, only cell culture studies. Thus, we recommend in a mouse model using humanized NSG mice (NOD/SCID/γ null, to evaluate the effects of α-GalCer on the cytotoxic activity of HuTCR-T1 γδ mice cell lines in vivo to better elucidate therapeutic potential and underlying mechanisms.
Author Response
Dear Reviewer # 2,
Thank you for the opportunity to respond to the comments on our manuscript. We appreciate the reviewers’ thorough evaluation and constructive feedback, which have helped us improve the quality of our work. We have addressed this point by clarifying details, a method, and revising sections. The relevant changes have been made in the manuscript. We believe these revisions have substantially strengthened the manuscript and hope it now meets the journal’s standards for publication. Thank you again for your time and consideration.
We have carefully addressed each point raised and summarize our responses below.
REVIEWER # 2
The report by Michel and colleagues presents the results from various human cancer cell lines co-cultured with and without α-GalCer, and humanized Humanized T cell receptor (HuTCR)-T1 mice gamma delta (γδ) T cells to explore whether γδ T expressing TCR can recognize CD1d–lipid complexes or become activated via lipid antigen presentation. They used standardized methodologies to verify the γδ T cytotoxic effects and possible biochemical mediators. Figures 1 and 2 show that HuTCR-T1 gamma delta (γδ) from various tissues from the mice exhibited greater cytotoxic effect compared to wild γδ T cells. Table 1 and 2 show that HuTCR-T1 mice γδ T cells after activation can express high the levels of various anticancer cytokines (IFN, TNF, IL-6, IL-17) and Granzyme B. Overall the results show that HuTCR-T1 γδ T mice model could be a good system for exploring the molecular mechanisms and possible clinical benefits of the adoptive transplantation of γδ T cells in patients. These aspects and other important considerations were described in the Discussion section of the report. One of the limitations of this study is that in vivo experiments were not performed, only cell culture studies. Thus, we recommend in a mouse model using humanized NSG mice (NOD/SCID/γ null, to evaluate the effects of α-GalCer on the cytotoxic activity of HuTCR-T1 γδ mice cell lines in vivo to better elucidate therapeutic potential and underlying mechanisms.
RE: We thank the reviewer for this insightful suggestion. While our current findings, based primarily on in vitro co-culture assays, provide valuable mechanistic insights into the cytotoxic potential of HuTCR-T1 γδ T cells, we acknowledge that in vivo studies are essential to fully evaluate their therapeutic efficacy, immune dynamics, and mechanistic effects in a physiological context.
However, in vivo studies could not be completed at this stage due to the ongoing breeding of the HuTCR-T1 mouse colonies and the need for optimization and approval of IACUC protocols for in vivo experimental designs. Future experiments using humanized NSG mice (NOD/SCID/γ null) will be planned to evaluate the in vivo cytotoxic activity of HuTCR-T1 γδ T cells following α-GalCer treatment. These studies will help further elucidate their therapeutic relevance and translational potential.
Reviewer 3 Report
Comments and Suggestions for Authors
The manuscript titled "The cytotoxic potential of humanized γδ T cells against human cancer cell lines in in vitro" explores the cytotoxicity of humanized γδ T cells (HuTCR-T1 mice) against various human cancer cell lines, demonstrating that these engineered cells show enhanced killing and cytokine secretion compared to WT γδ T cells, particularly under α-GalCer stimulation. The study uses LDH assays, PI staining, ELISPOT, and ELISA to assess cytotoxicity, apoptosis, and cytokine production, and confirms specificity using CD3e blockade. The use of humanized γδ TCR mice to assess human cancer cell line killing is innovative and provides a physiologically relevant model for preclinical immunotherapy studies. Combination of LDH, PI, ELISPOT, and ELISA provides converging evidence for enhanced cytotoxicity. Here are some comments:
1. In vivo validation is missing. While the in vitro results are compelling, the authors themselves acknowledge the limitations of in vitro systems in replicating the TME complexity. Including pilot in vivo data or explicitly outlining an in vivo experimental plan would strengthen the translational relevance.
2. Limited exploration of mechanistic pathways.Although granzyme B and cytokine production were measured, mechanistic pathways (e.g., FasL/TRAIL, death receptor pathways) are discussed but not tested. Functional blocking or transcript-level data (RT-qPCR or RNA-seq) could further clarify mechanisms. Dose it possible to explore?
Author Response
Dear Reviewer # 3,
Thank you for the opportunity to respond to the comments on our manuscript. We appreciate the reviewers’ thorough evaluation and constructive feedback, which have helped us improve the quality of our work. We have addressed this point by clarifying details, a method, revising sections. The relevant changes are made in the manuscript. We believe these revisions have substantially strengthened the manuscript and hope it now meets the journal’s standards for publication. Thank you again for your time and consideration.
We have carefully addressed each point raised and summarize our responses below.
REVIEWER # 3
The manuscript titled "The cytotoxic potential of humanized γδ T cells against human cancer cell lines in in vitro" explores the cytotoxicity of humanized γδ T cells (HuTCR-T1 mice) against various human cancer cell lines, demonstrating that these engineered cells show enhanced killing and cytokine secretion compared to WT γδ T cells, particularly under α-GalCer stimulation. The study uses LDH assays, PI staining, ELISPOT, and ELISA to assess cytotoxicity, apoptosis, and cytokine production, and confirms specificity using CD3e blockade. The use of humanized γδ TCR mice to assess human cancer cell line killing is innovative and provides a physiologically relevant model for preclinical immunotherapy studies. Combination of LDH, PI, ELISPOT, and ELISA provides converging evidence for enhanced cytotoxicity. Here are some comments:
1-In vivo validation is missing. While the in vitro results are compelling, the authors themselves acknowledge the limitations of in vitro systems in replicating the TME complexity. Including pilot in vivo data or explicitly outlining an in vivo experimental plan would strengthen the translational relevance.
RE: We appreciate the reviewer’s valuable comment regarding the absence of in vivo validation. These initial findings are based on in vitro co-culture assays. While these assays provide valuable mechanistic insights into the cytotoxic potential of HuTCR-T1 γδ T cells, we fully acknowledge that in vitro systems cannot fully recapitulate the complexity of the TME, and that in vivo studies are essential to validate and extend our findings.
At present, we are in the process of expanding and characterizing the HuTCR-T1 γδ mouse colonies and obtaining IACUC protocols approval for in vivo protocols necessary for such studies. Due to these logistical and technical constraints, we were unable to include in vivo data in the current study.
Nevertheless, we have now outlined a detailed in vivo experimental plan in the revised manuscript to emphasize the future translational direction. Specifically, we intend to utilize humanized NSG (NOD/SCID/γ null) mice to investigate the effects of α-GalCer on the cytotoxic activity of HuTCR-T1 γδ T cells in vivo, including tumor clearance, survival outcomes, and cytokine responses. We believe this addition strengthens the future outlook and translational value of our findings. (Lines 379).
- Limited exploration of mechanistic pathways. Although granzyme B and cytokine production were measured, mechanistic pathways (e.g., FasL/TRAIL, death receptor pathways) are discussed but not tested. Functional blocking or transcript-level data (RT-qPCR or RNA-seq) could further clarify mechanisms. Dose it possible to explore?
RE: We thank the reviewer for the valuable suggestion regarding the exploration of mechanistic cytotoxic pathways. Although Granzyme B expression and cytokine profiles were assessed, we fully agree that evaluating additional pathways such as FasL/Fas, TRAIL/DR4-5 signaling mechanisms would provide further insight into the effector functions of HuTCR-T1 γδ T cells.
At this stage, we are in the process of breeding a sufficient HuTCR-T1 γδ mouse colony and will need to optimize in vitro conditions for transcript-level analyses (e.g., RT-qPCR, RNA-seq) and functional blocking assays. Due to these ongoing efforts and logistical constraints, we were unable to incorporate this additional mechanistic data in the current study.